# Hydrogenase-based oxidative biocatalysis without oxygen

Ammar Al-Shameri [1], Dominik L. Siebert [1], Samuel Sutiono[1], Lars Lauterbach[2] & Volker Sieber [1,3,4,5] ✉

Biocatalysis-based synthesis can provide a sustainable and clean platform for producing chemicals. Many oxidative biocatalytic routes require the cofactor $NAD^+$ as an electron acceptor. To date, NADH oxidase (NOX) remains the most widely applied system for $NAD^+$ regeneration. However, its dependence on $O_2$ implies various technical challenges in terms of $O_2$ supply, solubility, and mass transfer. Here, we present the suitability of a $NAD^+$ regeneration system in vitro based on $H_2$ evolution. The efficiency of the hydrogenase-based system is demonstrated by integrating it into a multi-enzymatic cascade to produce ketoacids from sugars. The total $NAD^+$ recycled using the hydrogenase system outperforms NOX in all different setups reaching up to 44,000 mol per mol enzyme. This system proves to be scalable and superior to NOX in terms of technical simplicity, flexibility, and total output. Furthermore, the system produces only green $H_2$ as a by-product even in the presence of $O_2$.

The synthesis of fine chemicals via enzymatic routes offers a sustainable and clean platform compared to traditional organic chemical synthesis[1]. This field grew extensively in the past decades, chemicals railing from small alcohols and amines to complex pharmaceuticals were synthesized solely via enzymatic routes[2–6]. Most of these include at least one reduction and/or oxidation step that is catalyzed by single or multiple oxidoreductases within the route. Many oxidoreductases require electron donors/acceptors in the form of cofactors to catalyze their reactions[7].

In whole cells, cofactors are regenerated within the cellular metabolic pathways of the cell. Unlike living cells, cell-free enzymatic routes are easy to tune, they also circumvent possible cellular toxicity problems, side reactions, and metabolite competition. In addition, cell-free systems do not require extra mechanisms to transfer substrate or products via the cells' boundries[8,9]. When cell-free systems require a stoichiometric supply of cofactors, typically cofactor-regeneration systems are used, to overcome the high costs and low stability of cofactors. Various effective enzymatic, electro, and chemical regeneration systems have been established in the past years to regenerate

the reduced cofactor NADH[10]. Likewise, regeneration systems to recycle the oxidized cofactor $NAD^+$ (Fig. 1) have also been reported in recent years. For example, the reduction of pyruvate to lactate by lactate dehydrogenase is very efficient in this regard but the high cost of pyruvate limits its usage to a small scale. The same drawbacks are present in the α-ketoglutarate/glutamate dehydrogenase system. Whereas the toxicity of isopropanol and the high evaporation rate of acetone are major drawbacks of utilizing the acetone/isopropanol alcohol dehydrogenase system[11]. NADH oxidases (NOXs) remain up to date the most effective $NAD^+$ regeneration system in terms of activity, favorable thermodynamics, and low waste generation[12–14].

NOXs oxidize NADH using $O_2$ as a terminal electron acceptor. Depending on their type, NOXs generate either water or hydrogen peroxide as a by-product[15–17]. Similar to other oxidases, NOXs' dependency on $O_2$ makes their utilization for broad applications technically challenging[18]. The low transfer rate of $O_2$ from the gas into the liquid and the low solubility of $O_2$ especially at higher temperatures (0.26 mM at 25 °C) reduces the availability of $O_2$ in the reaction solutions[19]. Especially since many NOXs exhibit only a low affinity

[1]Chair of Chemistry of Biogenic Resources, TUM Campus Straubing for Biotechnology and Sustainability, Technical University of Munich, Schulgasse 16, 94315 Straubing, Germany. [2]RWTH Universität Aachen, Institute of Applied Microbiology, Worringerweg 1, 52074 Aachen, Germany. [3]Catalytic Research Center, Technical University of Munich, Ernst-Otto-Fischer-Straße 1, 85748 Garching, Germany. [4]SynBiofoundry@TUM, Technical University of Munich, Schulgasse 16, 94315 Straubing, Germany. [5]School of Chemistry and Molecular Biosciences, The University of Queensland, St. Lucia, QLD 4072, Australia. ✉e-mail: sieber@tum.de

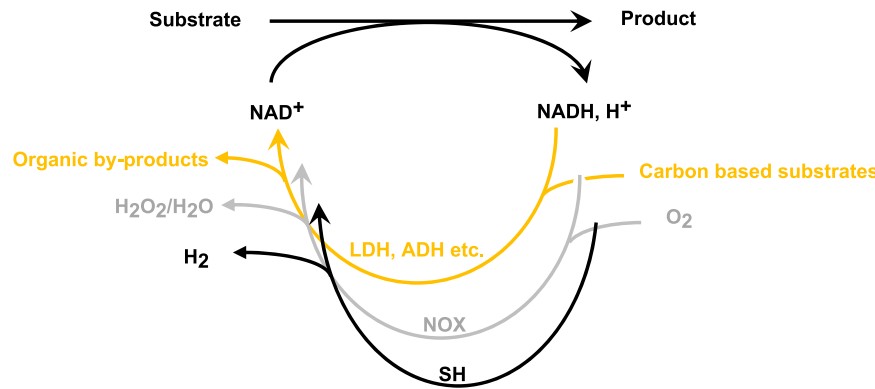

**Fig. 1 | Hydrogenase-based NAD⁺ regeneration system (black) compared to NADH oxidase (gray) and carbon-based systems (orange).** The hydrogenase-based system oxidizes NADH with 100% atom efficiency without additional oxidants and produces only $H_2$ as a clean by-product. Conventional NAD⁺ regeneration systems require additional oxidants as co-substrate and produce co-products. Water-producing NOX the cleanest and most efficient system up-to-date requires $O_2$ as a co-substrate to recycle NAD⁺. LDH lactate dehydrogenase, ADH alcohol dehydrogenase, NOX NADH oxidase, SH NAD⁺ reducing hydrogenase.

towards $O_2$ this requires elevating the $O_2$ pressure by aeration or even oxygenation to enhance the concentration of $O_2$ in the aqueous phase[20,21]. Aeration, however, commonly results in increased gas-liquid interfacial areas causing enzyme inactivation[22]. Specifically, the NOX variants that have been utilized so far show remarkably low process stability. Accordingly, NOX has to be constantly added in intervals throughout the reaction to achieve high conversion of substrate to product when applied under aeration[23].

Increasing the $O_2$ transfer rate into the aqueous phase represents a serious challenge in terms of energy supply and economic output at an industrial scale[24]. Therefore, NOX would not be the best choice for large-scale applications. In recent work by Bommarius's group, a volumetric oxygen transfer coefficient ($k_L a$) of 24 h⁻¹ was needed for a full ADH-NOX-catalyzed deracemization in a sparged bubble column[25]. Such $k_L a$ value will require a power input of at least 8000 W per unit volume at high flow rates that will cause inevitable enzyme inactivation. Reaching the same $k_L a$ value at slower flow rates (aeration) will require an even much higher power input[26]. An efficient $O_2$-independent system for NAD⁺ regeneration ideally will bypass these technical issues, will maintain enzyme integrity, and will be more desirable for a broad range of applications.

In recent years, the soluble hydrogenase (SH) from *Ralstonia eutropha* (also known as *Cupriavidus necator*) has been established as an effective regeneration system for reduced nicotinamide and flavin cofactors[27,28]. The $H_2$-driven cofactor reduction has the advantage of being 100% atom-efficient, non-carbon-based, and by-product free since it uses only $H_2$ as a reductant[29]. Furthermore, unlike the majority of hydrogenases, SH is $O_2$ tolerant; i.e. it retains its catalytic activity at ambient concentrations of $O_2$[30]. This made SH a successful system for recycling reduced nicotinamide and flavin cofactors in $O_2$-dependent multi-enzymatic cascades and biotranformations[28,31–33]. SH splits $H_2$ at the metal NiFe active site in the hydrogenase module, the electrons generated from $H_2$ oxidation are then transferred via Fe−S clusters and FMN molecules to NAD⁺ at the diaphorase module. NAD⁺ serves as a terminal electron acceptor and takes protons from the solutions via a prosthetic FMN to generate NADH[34]. The reverse electron transfer from NADH to $H_2$ generation has been also reported for SH in vivo and under specific conditions with low yields[35].

Exploiting the NADH oxidation activity of SH could offer a solid platform to regenerate NAD⁺ in an atom-efficient manner and circumvent the limitations related to using NOXs. In addition, generating $H_2$ as the only by-product will be useful for further $H_2$-driven biocatalytic applications.

In this work, we demonstrate an $O_2$-independent NAD⁺ regeneration system that generates $H_2$ as the only by-product by utilizing the NADH oxidation activity of SH. To validate the system, SH is coupled with NADH-dependent dehydrogenases in different setups, and the total output of NAD⁺ regeneration is tested and compared to NOX.

## Results and discussion

At the beginning, we tested the SH-mediated NADH oxidation and we could observe an activity of 0.22 U mg⁻¹ at 1 mM NADH. In earlier studies, it was shown that added FMN increases $H_2$-driven NAD⁺ reduction activity due to the loss of the endogenous FMN at the hydrogenase module during purification[36–39]. We observed that by adding FMN the activity of NADH oxidation increased by sevenfold (Supplementary Fig. 1a). The highest NADH oxidation activity for the isolated SH was determined at the concentrations of 1 mM and 0.1 mM for NADH and FMN respectively (Supplementary Table 1). As has been reported, SH can also utilize $H_2$ to reduce FMN to $FMNH_2$ which subsequently reacts rapidly with $O_2$ to $H_2O_2$[28]. Thus, we analyzed the formation of $H_2O_2$ during NADH oxidation using a colorimetric coupled assay with horseradish peroxidase but no $H_2O_2$ was detected in our samples.

We then tested if under these conditions (1 mM NADH and 0.1 FMN) the reverse electron flow will result in $H_2$ generation. We analyzed the $H_2$ evolution during the NADH oxidation with FMN using a $H_2$ microsensor. However, very minimal $H_2$ generation could be observed under these conditions (Supplementary Fig. 1b), presumably due to the low amounts of NADH oxidized in the absence of an NAD⁺ consuming system as the equilibrium constant of this reaction is ca. $2*10^{-7}$ and the standard redox potentials for $2H^+/H_2$ and NAD⁺/NADH are -413 and −320 mV, respectively[40,41].

After optimizing the SH oxidation activity and confirming that no $H_2O_2$ was formed, we wanted to examine the competence of this system in substituting NOX for the regeneration of NAD⁺. Therefore, we examined the SH NAD⁺-regeneration system in both single-step biotransformation and a multi-enzymatic cascade and compared SH performance to a NOX-based NAD⁺ regeneration.

### Applicability in single-step biotransformation

As single-step biotransformation, we chose the oxidation of D-xylose to D-xylonate by the xylose dehydrogenase from *Herbaspirillum seropedicae* (HsXylDH2). HsXylDH2 oxidizes D-xylose using NAD⁺ as the hydride acceptor to generate NADH, thus, the catalytic activity of HsXylDH2 is limited by the availability of NAD⁺. We coupled SH with HsXylDH2 at different setups and compared it to the same biotransformation applied with a FAD-dependent NADH oxidase from *Lactobacillus pentosus*. As comparing criteria, we analyzed the final production yield of D-xylonate and the total turnover of the regeneration system (TTN) in both systems. We decided to use Tris-HCl pH 8

and 30 °C as our conditions when comparing both systems to eliminate any possible biases (Supplementary Table 2).

First, we tested the effect of $O_2$ on the yield of the biotransformation with SH. No significant difference was observed in the samples with $O_2$ even the one containing catalase (Supplementary Table 3, entry 1-3). This confirmed our earlier observation that no $H_2O_2$ was formed in the presence of $O_2$ and that the system was $O_2$ stable. We still added catalase to all of our upcoming reactions as a precaution. Next, we tested the formation of D-xylonate from D-xylose in both systems using equi-unit amounts of SH and NOX. The conversion rate of the substrate decreased drastically after the first minute in both

systems. While the conversion rate of the substrate in SH samples remained nearly constant at around 10 μmol h⁻¹ in the first hour, it was almost zero in NOX samples for the rest of the reaction (Fig. 2). Additionally, the total product formation in SH samples was fourfold higher than in NOX samples. This demonstrates that the availability of $O_2$ is the bottleneck in the NOX-catalyzed NADH oxidation and that releasing $H_2$ from the system is more feasible than dissolving $O_2$.

## Applicability in multi-enzymatic cascades

Our next step was to validate the SH regeneration system for the recycling of NAD⁺ in multi-enzymatic cascades. We have chosen our recently developed cascade that produces α-ketoglutarate (α-KG) from D-xylose since it involves two NAD⁺-dependent enzymes and therefore two NAD⁺ regenerations steps are required (Fig. 3)[42]. Due to the high amount of NAD⁺ required to run this cascade, we tested how far removing $H_2$ from the system will influence the yield of the conversion. Since the oxidation of NADH towards $H_2$ synthesis by SH is endergonic but entropy-driven, removing the by-product ($H_2$) will push the reaction towards NADH oxidation. Removing $H_2$ from the system was achieved by opening the reaction vessel, and indeed resulted in a 45% increase in both the conversion and total turnover number of SH (Supplementary Fig. 2).

Another important factor that has to be considered when applying a cofactor-regeneration system in cascades is the effect of the cofactor concentration on product formation. This may differ from the determined kinetic parameters for the regeneration system with isolated enzymes. Unlike with isolated SH, we found that increasing the NAD⁺ concentration to 10 mM significantly enhanced product formation even at low SH concentration (Table 1, and Supplementary Fig. 3). Without a rapid and constant NAD⁺ consumption, NAD⁺ will be reconverted to NADH by SH using the dissolved $H_2$. This might explain the low NADH oxidation activity at higher NADH concentrations with isolated SH.

A full conversion of D-xylose to α-ketoglutarate was achieved in both systems with 100 mM of D-xylose (Table 1, entries 4, and 8). However, full conversion with NOX was only possible when an excessive amount of NOX was used, here around 70 μM. The same was reported by Beer et al. who showed that the initial concentration of NOX had a significant impact on the final product yield, which is due to the low stability of NOX with a half-life of only approx. 3 h at 37 °C[13,15,23]. The space-time yield of the SH system reached up

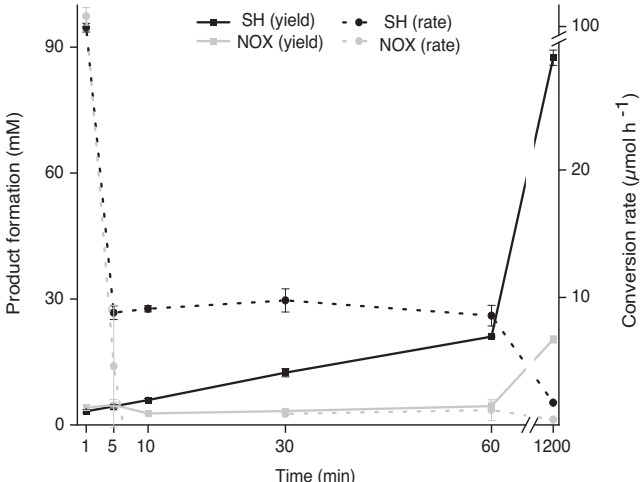

**Fig. 2 | The product formation (solid) and conversion rate (dashed) in the conversion of D-xylose to D-xylonate with both regeneration systems SH (black) and NOX (gray) over time.** Reaction conditions: 500 μL of 1 M Tris-HCl pH 8 with 0.5 M D-xylose, 10 mM NADH, and 0.1 mM FMN, at 30 °C and 550 rpm. Equi-unit (3 U mL⁻¹) of each SH (black) and NOX (gray) were added to the reaction. This corresponds to 4.5 μM and 19.5 μM for NOX and SH, respectively. Both *HsXyl*DH2 and *Nm*lac2 (lactonase) were added in excessive amounts (60 μM and 4 μM for *HsXyl*DH2 and *Nm*lac2, respectively) to drive the reaction towards D-xylonate. SH soluble hydrogenase, NOX NADH oxidase. Each reaction with the same conditions was conducted in independent replicates (*n* = 2, biologically independent). The data are depicted as mean values with error bars as SD.

**Fig. 3 | Cell-free synthetic cascade reaction of biotransformation of D-xylose to α-ketoglutarate.** The enzymes that require NAD⁺ are highlighted in pink. The hydrogenase-based regeneration system (SH) and $H_2$ are highlighted in bold black. $H_2$ is the only by-product of NAD⁺ regeneration by SH. The $O_2$-dependent NOX system is highlighted in gray and is used as a control. SH soluble hydrogenase, NOX NADH oxidase, HSXylDH2 xylose dehydrogenase, NmLac lactonase, PuDHT Xylonate dehydratase, KdpD 2-keto-3-deoxy-D-xylonate dehydratase, KgsaDH α-ketoglutarate semialdehyde dehydrogenase.

to ~28 mmol L$^{-1}$ h$^{-1}$ after 3 h and ~16 mmol L$^{-1}$ h$^{-1}$ after 16 h (Supplementary Fig. 3). This yield exceeds most of the space-time yields reported for NADH oxidases in vitro systems and matches the data reported for immobilized whole cells co-expressing NOX[13,43,44].

## Performance of SH vs. NOX at different setups

Next, the performance of both regeneration systems was compared at different technical setups using the same amount of both enzymes. Despite its lower NADH oxidation-specific activity compared to NOX, SH outperformed NOX shows in all setups as Fig. 4. Removing H$_2$ from the system was much more favorable than transferring O$_2$ into the system. This is expected since degassing is entropically more favorable than dissolving gases in aqueous solutions. As expected, the shaking speed correlated directly with the amount of NAD$^+$ recycled. Even without shaking the removal of H$_2$ was possible but this setup was three times less productive than at high shaking speed. For NOX increasing the shaking speed and opening the system had only a small effect. Increasing O$_2$ content will require an active transfer of O$_2$ into

the system by bubbling[23]. However, here NOX had to be added every 30 mins due to the enzyme inactivation induced by bubbling. Another alternative is to increase the reaction surface by increasing the vessel volume and the head space so that the catalysis could take place on the gas-liquid interface.

Both options seem to be feasible on a large scale however they represent real challenges in terms of space availability and energy consumption. Removing H$_2$ on the other hand seems to be readily achievable using conventional stirring, which is widely applied in large-scale processes for mixing. We tested how both systems perform in stirred open vessels. Here as well, TTN of SH outperformed NOX despite the decline in the stability of all enzymes during stirring (Supplementary Fig. 4). Commonly, most isolated enzymes lose their stability due to the shear forces resulting from stirring. The mechanical enzyme stability could be easily enhanced by immobilization[45].

## H$_2$ evolution

To analyze the fate of the electrons released from NADH oxidation, the H$_2$ evolution was followed during the enzymatic biotransformation using a H$_2$ sensor in the aqueous phase. A rapid H$_2$ production was observed in the first 10 mins of the conversion of D-xylose to D-xylonate reaching a volumetric rate of up to 1 mmol L$^{-1}$ h$^{-1}$ and an integrated yield of H$_2$ of 2.5 mM in the aqueous phase (Fig. 5a). This result confirmed the reverse electron transfer from NADH to the NiFe active center resulting in H$_2$ production under these conditions. After a rapid increase in the H$_2$ concentration and an equilibrium phase for almost 2 hours the H$_2$ concentration in the aqueous phase gradually decreased. This is due to the very low solubility of H$_2$ in water (max. 0.78 mM at 25 °C) and the constant diffusion of H$_2$ into the gas phase. This also explains the huge difference between the rate of product formation (Fig. 2) and the rate of H$_2$ evolution in the aqueous phase.

Adding pulses of D-xylose pushed the system towards more H$_2$ production. This might be related to a possible reverse reaction of HsXylDH2 without lactonase towards D-xylose from D-xylonolactone, which decreased the available NADH. The H$_2$ production yield and rate were increased at higher NAD$^+$/NADH concentrations. In addition, the H$_2$ content in the aqueous phase decreased upon stirring the reaction, which enhances the release of H$_2$ into the gas phase (Supplementary Figs. 5–7).

High production of H$_2$ was also observed in the conversion of D-xylose to α-ketoglutarate and was correlated with the highest TTN (4.4 × 10$^4$) for SH for NADH oxidation so far (Table 1 entry 5 and Fig. 5b). The integrated yield of H$_2$ in the aqueous phase in the open system was around 4.2 mM, which is only 4% of the product yield and which demonstrates its effective removal into the gas phase. Determining the whole H$_2$ produced in the open system was very

### Table 1 | The conversion of D-xylose to α-ketoglutarate after 16 h

|  | Entry | SH/NOX (µM) | NAD$^+$(mM) | α-ketoglutarate(%) | D-xylonate (%) |
|---|---|---|---|---|---|
| SH | 1[a] | 70 | 1 | 2 | 39 |
|  | 2[a] | 20 | 10 | 14 | 23 |
|  | 3 | 39 | 10 | 100 | 0 |
|  | 4[b] | 30 | 10 | 100 | 0 |
|  | 5[c] | 5 | 10 | 75 | 0 |
| NOX | 6[a] | 4 | 1 | 1.2 | 0.8 |
|  | 7[a] | 58 | 10 | 25 | 0 |
|  | 8 | 72 | 10 | 100 | 0 |
|  | 9[b] | 77 | 10 | 52 | 0 |
|  | 10[c] | 25 | 10 | 16 | 0 |

Reaction conditions: 500 µL of 1 M Tris-HCl pH 8 containing 0.1 M D-xylose, 1 mM MgCl$_2$, and 0.1 mM FMN in 10 mL deep well plates sealed with a breathable membrane. The reaction was conducted at 30 °C and 550 rpm.
All other enzymes were added as stated in the experimental section. Each reaction with the same conditions was conducted in independent replicates (n = 2, biologically independent). For all other enzymes were added as stated in the experimental section.
[a]0.5 M D-xylose.
[b]1.5 mL tubes with pierced lids, shaking at 120 rpm.
[c]Unstirred at 22 °C, 0.15 M D-xylose.

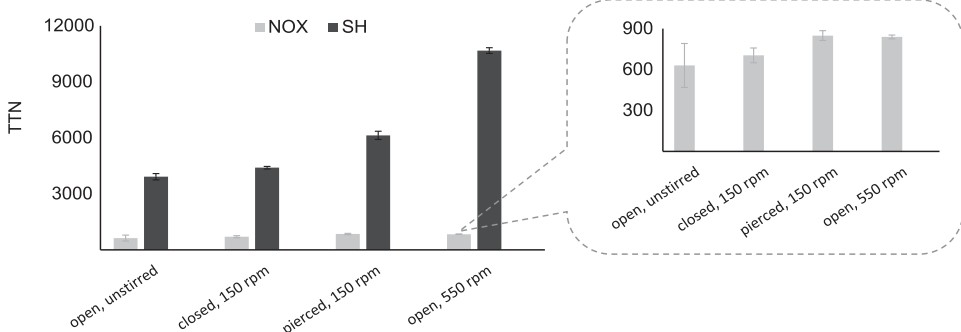

**Fig. 4 | The total turnover number (TTN) of SH (black bars) and NOX (gray bars) during the conversion of D-xylose to α-ketoglutarate at different setups.** The TTN was calculated as the final total NAD$^+$ recycled in the system per enzyme unit during the whole reaction. All reactions were performed in 400 µL reaction buffer 1 M Tris-HCl pH 8, with equal amounts of enzymes (7.8 µM), 0.5 M D-xylose, 1 mM MgCl$_2$, 0.1 mM FMN, 10 mM NADH at 30 °C. The closed systems contain 1 mL of head space. For all other enzymes were added as stated in the experimental section. SH soluble hydrogenase, NOX NADH oxidase each reaction with the same conditions was conducted in independent replicates (n = 2, biologically independent). The data are depicted as mean values with error bars as SD.

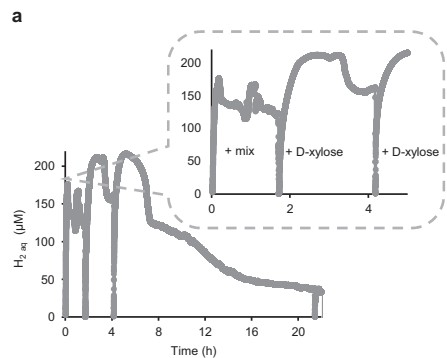

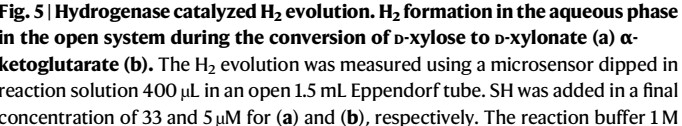

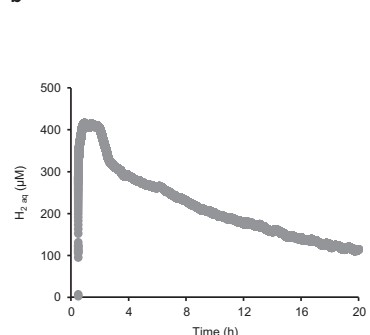

**Fig. 5 | Hydrogenase catalyzed H₂ evolution. H₂ formation in the aqueous phase in the open system during the conversion of D-xylose to D-xylonate (a) α-ketoglutarate (b).** The H₂ evolution was measured using a microsensor dipped in reaction solution 400 μL in an open 1.5 mL Eppendorf tube. SH was added in a final concentration of 33 and 5 μM for (**a**) and (**b**), respectively. The reaction buffer 1 M Tris-HCl pH 8 contained 0.1 M D-xylose, 10 mM NADH, and 0.1 mM FMN and was performed without stirring at 30 °C (**a**) and 22 °C (**b**). +mix: short stirring for 15 seconds at 500 rpm. +D-xylose: addition of 20 mM of D-xylose. For the conversion of D-xylose to D-xylonate 30 μM of HsXylDH2 was used. For the α-ketoglutarate cascade, all other enzymes were added as stated in the experimental section.

challenging due to the constant diffusion of H₂ into the gas phase (Supplementary Fig. 8), we discuss this issue in details in the supporting information.

## Upscaling

Finally, we aimed to test whether the SH system can be scaled up. However, producing sufficient amounts of SH in the main host (*Ralstonia eutropha*) takes ~7–10 days and is not compatible with standardized cultivation conditions and media (glucose, LB, and TB). This is a big drawback when scaling up is intended, therefore, we worked on optimizing the production of SH in *E. coli*. For the expression of SH in *E.coli*, we used a similar system to the one reported by Schiffels et al.[46]. We were able to produce sufficient amounts of functional SH using standard cultivation media and conditions. The activity of SH produced in *E. coli* was almost identical to the one produced in the native host. The amount of protein produced in one-liter culture per day exceeded this of in the native host and the yield reported in *E. coli* (Supplementary Table 4)[46]. This might be related to the strep-tagged HoxI that dominated the protein sample in Schiffels et al, despite its insignificance for the functionality of SH.

Producing functional hydrogenases on a large scale is generally challenging due to the lack of maturation machinery required to build the metal cofactor of most hydrogenases in *E. coli*. Cloning the maturation machinery in *E. coli* can be a good strategy to solve this issue. Even more promising is in vitro reconstitution, here, the apoprotein is produced in cells and the metal cofactor is synthesized chemically and incorporated into the apoprotein upon purification to give a functional enzyme[47,48]. Theoretically, both the apoprotein and the metal cofactor can be easily produced in large amounts.

After successfully producing SH in *E. coli*, we scaled up our reaction by 25-fold and performed the reaction on a 10 mL scale, the amount of all other enzymes was also scaled up by 20-fold, and a final concentration of 7 μM of SH was used. On the 0.4 mL scale the same amount of SH resulted in the formation of 41 mM of α-ketoglutarate and a TTN of ~10,000 (Fig. 4). As a preliminary strategy to approach industrial processes, we replaced shaking with stirring and contentiously removed H₂ from the system. This was achieved by setting up a consistent flow of N₂ into the gas phase of the reaction similar to the setup of del Campo et al.[49]. Both procedures belong to standard operations in the industry.

The 10 mL setup enabled us to measure the H₂ in the gas phase at the exhaust which was not possible at a 0.4 mL scale. We could successfully produce 28–37 mM of product with a TTN above 8000–9000 in 44 h (Supplementary Table 5, entry 1–2). This is ~90% of the yield obtained in a 0.4 mL scale which proves that the system is linearly scalable. The cumulative yield of H₂ from the gas phase reached around 205 μmol after 24 h, which is 63% of the product yield (Fig. 6). The other 37% are most probably lost due to diffusion since the setup was built with standard laboratory equipments that are not H₂ tight. This explains also why after 44 h the H₂ yield was only 45% of the product yield. By further scaling up the system to 40 mL using just 1 μM of SH we could demonstrate the functionality of the system with minimal amounts of enzyme achieving TTN as high as 17000 without stirring (Supplementary Table 5, entry 3). The impact of high-speed stirring on enzyme stability became evident (Supplementary Table 5, entry 4). This suggests that it might be important to add the enzymes in portions to minimize the destabilization effect caused by stirring.

Tuning the system towards higher H₂ production will require further optimizations in terms of reaction setup and using sophisticated materials to ensure better H₂ capture. To overcome the safety issues related to H₂ handling, the produced H₂ can be used as an electron donor in electrochemical applications or to fuel reductive reactions in connected chemical synthesis[31,50].

In conclusion, we presented a efficient hydrogenase-based concept for the regeneration of NAD⁺. The hydrogenase-based system demonstrated its competence to recycle NAD⁺ in enzymatic cascades and sustain a full conversion of substrate to the product reaching a high TTN. Hydrogenase produces only H₂ as a by-product, which can be easily removed from the system and further used as a clean and sustainable energy carrier to drive other biocatalytic reactions[28,31,32]. The hydrogenase-based system proved to be superior to the conventional system of NADH oxidase in terms of efficiency, TTN, and technical setup. Furthermore, removing H₂ from the reaction proved to be more favorable and much easier than transferring O₂ into the reaction. Since the hydrogenase system is O₂-independent, it can be used under anaerobic conditions if the substrates or products are O₂-sensitive. Unlike NOX, the hydrogenase system does not require an extra supply of O₂ by sparging or bubbling, which leads mostly to enzyme inactivation and post challenges in terms of energy supply, and expenses. Therefore, the hydrogenase system offers more flexibility and tunability (oxidation/reduction) compared to NOX. Finally, the O₂ tolerance of SH enables its coupling with other O₂-dependent enzymes like oxidases and monooxygenases in explosion-safe setups. Such a coupling will be very challenging with NOX since these enzymes will compete with NOX for O₂ and an excessive supply of O₂ will be necessary. The production of H₂ in the presence of O₂ with such high yields is up to our knowledge still unknown. This opens the door for many future bio-economical applications and green energy technologies such as light-driven H₂ production. Such technologies have a huge potential to contribute to the future green economy.

a

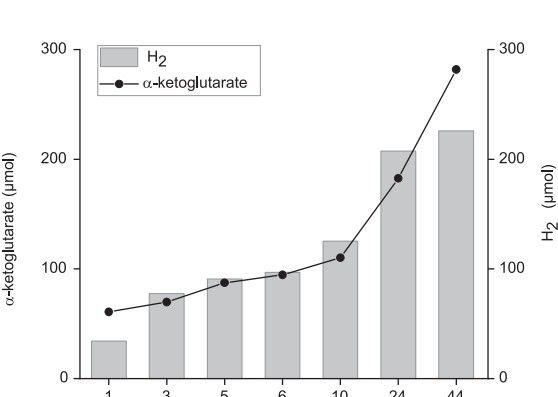

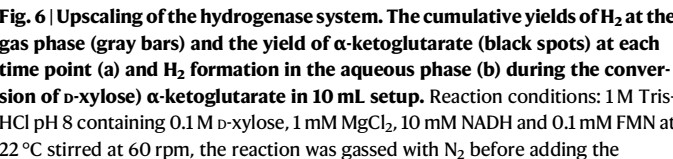

b

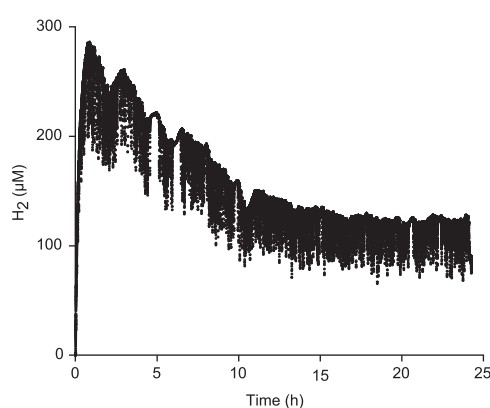

**Fig. 6 | Upscaling of the hydrogenase system. The cumulative yields of H₂ at the gas phase (gray bars) and the yield of α-ketoglutarate (black spots) at each time point (a) and H₂ formation in the aqueous phase (b) during the conversion of D-xylose) α-ketoglutarate in 10 mL setup.** Reaction conditions: 1 M Tris-HCl pH 8 containing 0.1 M D-xylose, 1 mM $MgCl_2$, 10 mM NADH and 0.1 mM FMN at 22 °C stirred at 60 rpm, the reaction was gassed with $N_2$ before adding the enzymes. All other enzymes were added as stated in the experimental section and 7 μM of SH were added. The reaction was performed under a flow of $N_2$ (2.4–2.8 mL min⁻¹) in a glass flask with a gas inlet and outlet. The gas at the exhaust was analyzed via GC-TCD to determine the amount of H₂. The production of 1 mol of α-ketoglutarate will give rise to 2 mol of H₂.

## Methods

### Expression and protein purification

All enzymes involved in the cascade except SH were expressed and purified as reported by Sutiono et al.[42]. SH was expressed and purified as described by Lauterbach et al.[30]. The reconstitution of FAD into NOX was performed as described by Nowak et al.[15].

To express SH in *E.coli*, the auxiliary genes, and structural genes for SH were cloned as operons as described by Lamont et al.[51]. The structural genes without HoxI were inserted into a T7 expression system similar to the one reported by Schiffels et al.[46]. *E. coli* BL21-cells were transformed with the corresponding plasmids and cultivated on agar plates with suitable antibiotics. These plates were incubated at 37 °C overnight. One of the grown clones was transferred to LB-media with the same suitable antibiotics and grown overnight at 37 °C. This overnight culture was then used to inoculate a main culture in TB media containing 100 μM of both $NiCl_2$ and $FeCl_3$. These cultures (1 L) were grown at 37 °C in 2 L Thomson Ultra Yield® flasks until an OD of at least 2 was reached. Subsequently, IPTG was added to a final concentration of 1 mM, and the culture was transferred to room temperature, grown overnight, and harvested the following day.

### Protein concentration and TTN determination

Protein concentration was determined using a BCA assay (SERVA, Germany). The total turnover number is the ratio of the total amount of the yield product in μM to the total amount of enzyme applied in μM.

### Biotransformation

Before biotransformation residual glycerol, Na⁺, and K⁺ salts were removed by performing buffer exchange with 400 mM Tris-HCl pH 8 using Vivaspin filters (Sartorius, Switzerland). The biotransformations were conducted as described in the main text. The reactions were stopped by filtering the enzymes using 10 kDa spin filters (Sartorius, Switzerland). If not stated otherwise, all enzymes in the α-ketoglutarate cascade except SH and NOX were added depending on their activities in concentrations of (76, 6.8, 32, 28, 47, 3) μM for (HsXylDH2, NmLac2, PpD-KdpD, PuDHT, PpKgsaDH, Catalase), respectively.

### Hydrogen measurement

Hydrogen was measured in the aqueous phase using an NP hydrogen sensor (Unisense, Denmark). The sensor was calibrated using an H₂ saturated solution for 100% and $N_2$ saturated solution for 0%. H₂ was produced by the reaction of 1 M of $H_2SO_4$ with Zn. The software Sensorstrace suite was used for data analysis and monitoring.

For analyzing H₂ in the gas phase, a gas chromatography GC-TCD model Clarus 580 (Perkin Elmer, Germany) was used. Totalchrom (Perkin Elmer) was used to evaluate the data and integrate the chromatograms.

The GC is equipped with a thermal conductivity detector (TCD) for small gases, and Hayesep N column, and a molecular sieve column. Argon is used as the carrier gas. 20 mL of sample were taken from the exhaust, injected into the GC using a glass syringe, and analyzed. For the samples, in the first 10 h gas was collected directly into a mechanical glass syringe and analyzed via GC-TCD. For the overnight samples, gas samples were taken directly from the gas phase above the reaction, and the amount of H₂ was calculated taking into consideration the total volume of $N_2$ flowed during this time. The calibration of GC was conducted using air, 100% H₂, and a predefined commercial gas mixture.

### HPLC analysis

The yield of each biotransformation of D-xylose to D-xylonate and α-ketoglutarate was measured by HPLC coupled with UV and RI detectors. Before HPLC analysis, samples were prepared by diluting samples first in water (1:10), filtering them with a spin filter, then diluting the samples (1:10) in 2.5 mM $H_2SO_4$, and 10-20 μL was injected into HPLC. The HPLC program was set up as reported by Sutiono et al.[42]. In brief, the compounds were separated using an ion-exclusion column (Rezex ROA-Organic Acid H⁺ (8%), Phenomenex, Germany), run isocratically with $H_2SO_4$ 2.5 mM at 70 °C for 20 min. It is worth mentioning that due to evaporation that occurs as a result of opening the system, the concentration of the product was increased. This was considered when calculating the total conversion of the substrate. The chromatograms were analyzed and integrated using CHROMELEON®6.80 SR15 software.

### Reporting summary

Further information on research design is available in the Nature Portfolio Reporting Summary linked to this article.

## Data availability

The authors declare that the data supporting the findings of this study are available within the article and the Supplementary Information.

The raw data of HPLC, GC, sensor, spectrophotometer, calibrations, and calculations are provided in the source data file with this paper. All other data are available from the authors upon request. Source data are provided with this paper.

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

## Acknowledgements
A.A. and D.L.S. thank the Bundesministerium für Bildung und Forschung (BMPF), Project REDEFINEH2 (FZ: 01DD21005) for the financial support. This work was supported by the Deutsche Forschungsgemeinschaft (DFG) through the cluster of excellence EXC 2186—390919832—"The Fuel Science Center" (to L.L.). We thank Lena Würstl for the purification and preparation of NOX. We also thank Broder Rühmann and Anja Schmidt for their support during the HPLC analysis. We are very grateful to Professor Frank Sargent for his support in optimizing the production of SH in *E. coli*.

## Author contributions
A.A. performed all experiments, analyzed the results, and wrote the manuscript. D.S. produced and optimized the production of SH in E.coli. S.S. purified the enzymes and established the analysis protocols, L.L. contributed to the design of experiments and evaluation of data. V.S. developed the original concept and guided the outline of the study. All Authors contributed to writing and evaluating the manuscript.

## Funding

## Competing interests
The authors declare no competing interests.
