## [Peer Review File · Nature Communications]

Hydrogenase-based oxidative biocatalysis without oxygenREVIEWER COMMENTS

Reviewer #1 (Remarks to the Author):

Sieber and colleagues report on an enzymatic tandem system for the regeneration of NAD⁺ over NADH, a common biological cofactor. Commonly, NADH is oxidized en route reduction of O₂ by the enzyme NADH oxidase (NOX), which causes two main issues: (i) NOX is damaged by reactive oxygen species (as a side product of O₂ reduction), and (ii) O₂ supply demands purging the reaction vessel causing protein denaturation at the gas/solvent interface.

The authors now demonstrate that NADH oxidation can be coupled to proton reduction and H₂ evolution, replacing NOX against the 'soluble [NiFe]-hydrogenase' (SH) from *C. necator*. Technically, removing H₂ is far less a challenge than introducing O₂, in particular at elevated temperatures that will enhance catalytic activity. Furthermore, SH is fairly O₂-tolerant (unlike most hydrogenases), so this concept makes sense and will justify publication in Nature Communications.

Unfortunately, the manuscript lacks a sufficiently convincing data set, describing promising but eventually 'episodic' experiments instead of covering the whole parameter space. I would at least expect variations in pH and buffer type, as well as protein and salt concentration. Has detergent been used (NOX is a membrane-bound protein)? With respect to technological aspects, open, closed, and pieced vessels are compared, but the influence of stirring/shaking is not followed up upon. More critical maybe, no systematic variation of temperature is reported, and no approach in upscaling was conducted. Overall, the reported statistics are weak. Potential bottlenecks in the (industrial) production of hydrogenase are not discussed.

Pitching SH as an alternative to NOX in the context of biotechnology, I would expect a serious engineering-type analysis. In its current 'proof of concept' status, the manuscript may be suitable for Scientific Reports. However, I will be willing to review a re-submission to Nature Communications if the authors deliver the suggested data and statistics.

Reviewer #2 (Remarks to the Author):

Regeneration of enzyme cofactor is interesting work in enzymatic catalysis. In this manuscript, the authors used the soluble hydrogenase (SH) from *Ralstonia eutropha* to regenerate NAD⁺ which does not need oxygen. And the comparison of the NAD⁺ regeneration with NADH oxidase (NOX) was performed. The results suggested that SH has higher total turnover numbers of 40000 and only H₂ as a by-product compared with NOX. From this point of view, SH has advantages in regenerating NAD⁺. However, some issues should be more clearly and revised before publication. The detailed comments were listed as follows:

- 1) The title should be more specific and include hydrogenase, which is the key enzyme for the regeneration of NAD⁺.
- 2) When multiple enzymes are used together, pH compatibility is an important factor for the enzymatic reactions. Actually, the pH and buffer concentration of reaction conditions were 8.0 and 500 mM, which were optimal for SH. But they are not optimal conditions for HsXylDH₂ and NOX. From the previous publications, the optimal pH of most NADH oxidases falls into the pH range of 5.0-6.0. The evaluation of NOX under the optimal conditions of SH is not fair to NOX.
- 3) The specific activities of the NOX and SH are 50.1 U mg⁻¹ in 50 mM Tris-HCl pH 7.0 at 37°C and 1.5 U mg⁻¹ in Tris-HCl pH 8 at 22 °C, respectively. SH shows a quite low activity which means the low catalytic efficiency of the enzyme. However, the results showed that total turnover number of SH was much higher than that of NOX. Please give more explanation in the text.
- 4) In the experiments, equi-unit enzymes of NOX and SH were used for the biocatalysis. The protein concentration of the enzyme will be highly different, and the protein concentration of SH will be 33-fold that of NOX in the reaction mixture. This might have some effects on the enzymatic reactions because of the low concentration of NOX.
- 5) The total turnover number of NOX from *Lactobacillus pentosus* is 6.8*10⁵ in the previous reports. It is totally different to the data in this work. Please give the possible reasons.
- 6) It might be better to add a description of SH stability in utilized conditions for comparison because the H₂ in the reaction mixture might have negative effects on SH and other enzymes.
- 7) It is noted that the enzymatic reactions were carried out in the close and open systems. The yield in the open system was 45% higher than that of the close one. Thus, it is necessary to evaluate the effects of the pressure and the solubility of H₂ in the system.

Reviewer #3 (Remarks to the Author):

In their manuscript 'Oxidative Biocatalysis without Oxygen' Sieber and coworkers demonstrate the feasibility of hydrogenase-catalysed NAD⁺ regeneration to drive dehydrogenase-catalysed oxidation reactions. Particularly attractive about this new NAD⁺ regeneration system is the volatility of the H₂ coproduct thereby shifting thermodynamically unfavourable oxidations reactions and the independence from a gaseous or organic cosubstrate.

To the best of my knowledge this approach is conceptually novel thereby principally qualifying this manuscript for publication in Nature Communications. I have no doubts about the quality of the experimental setups, experimental work and analysis of the data reported in this contribution. Nevertheless, I believe that the manuscript would profit from some extra experiments and corrections detailed below:

The authors state the thermodynamically beneficial irreversibility of the NAD⁺ regeneration reaction, which I agree with. However, so far the authors have used aldehyde to acid oxidation reactions, which intrinsically are irreversible themselves. It would be very interesting to see if the hydrogenase-system can also yield full conversion in challenging reaction systems such as the oxidation of primary or secondary alcohols to the corresponding carbonyl products. I am thinking about simple oxidations such as cyclohexanol or benzyl alcohol, which are typically difficult to drive to completion using substrate-coupled regeneration approaches.

Minor issues:

Line 36ff: The authors state that there are few effective NAD⁺ regeneration systems available and later (arbitrarily) mention some older systems. One could argue here that other NAD(P)⁺ regeneration systems have been reported in the past years. I am also not a big fan of the term 'efficient' per se if it is not quantified (TNs, TFs, STY etc.) and compared.

Line 70: I am not convinced that the hydrogenase-cased oxidation system is '100% atom efficient'. This is the case if reduction reactions are considered (then all atoms of the starting material are incorporated into the product). Here, H₂ as by-product is formed, hence the atom efficiency (as defined by Trost) is not perfect.

Line 78: I would not call it NADH oxidase but rather NADH oxidation activity. The term oxidase sounds too much like an O₂-dependent reaction.

Line 92: The authors state that no H₂O₂ was detectable but it remains unclear why. Were these reactions performed anaerobically? Or was there some catalase-activity present in the enzyme preparation? Any other explanation?

Line 116: 'The conversion rate of the substrate decreased drastically after the first minute...': As far as I can see the very high initial product formation rate is obtained from the first data point after one minute. I do not see a zero-value. So could the very high initial product formation rate be an artefact from this first value (maybe the t=0-value would not be zero?) If this is not the reason, the authors should provide some explanation for this observation.

Figure 1: Please provide molar concentrations of all reagents (here of the enzymes). This counts for all figures and tables. Please use colour to make identification of the individual time course easier.

Line 118: 10 μmol·h⁻¹: I assume the authors mean concentrations rather than absolute amounts here.

Line 128: rather than 'thermodynamically unfavorable' better 'endergonic'. Thermodynamic feasibility consists of enthalpic and entropic terms.

Table 1: better add the enzyme concentrations in the table rather than using the complicated +, ++, +++ scheme. Conversion column: this is a bit unclear. Maybe use conversion and D-xylonate/a-KG-selectivity or display the concentrations of the products (preferred).

Response to reviewers

Reviewer #1 (Remarks to the Author):

I would at least expect variations in pH and buffer type, as well as protein and salt concentration. Has detergent been used (NOX is a membrane-bound protein)? With respect to technological aspects, open, closed, and pieced vessels are compared, but the influence of stirring/shaking is not followed up upon. More critical maybe, no systematic variation of temperature is reported, and no approach in upscaling was conducted. Overall, the reported statistics are weak. Potential bottlenecks in the (industrial) production of hydrogenase are not discussed.

1. We thank the reviewer for his valuable suggestions and constructive criticism. In fact, the optimal pH, temperatures, salt effects and buffer types for both systems are already known in the literature. This also applies for the most of enzymes of the cascades. Since we are using wild-type enzymes, we do not expect this to change. Nevertheless, we were intrigued by the reviewer's suggestions and performed a screening to test the effect of pH, buffer and temperature on both systems; we added the new data to supporting information.
2. The NOX that we use is not a membrane-bound protein. We express this enzyme in *E.coli* and purify it from the soluble extract. All previous reported studies for the NADH oxidase of *Lactobacillus pentosus* showed that LpNOX is a cytoplasmic protein.
3. We did compare both systems with stirring Figure 4S and discussed that in main text. Now we added more information to Figure 4S.
4. Concerning the temperature, both enzymes operate best at around 30°C and both systems suffer from thermal instability in a very similar manner. Increasing the thermal stability of both enzymes by protein engineering is the focus of future work.
5. In regard to upscaling: we do agree 100 % with the reviewer in this point. We added an extra segment for the upscaling. We scaled up the reaction 20fold and 80fold and discussed the issues related to that in light of enzyme stability, stirring and H₂ analysis.
6. We also agree with the review about the issues related to production of hydrogenases. Recombinant production of Hydrogenase in *E.coli* has been reported and in our additional experiments, we have employed this approach. A full segment is added discussing this point.

Reviewer #2 (Remarks to the Author):

- 1) The title should be more specific and include hydrogenase, which is the key enzyme for the regeneration of NAD⁺.

We do agree with the reviewer. This has been revised. The title has been changed to include hydrogenase in it.

- 2) When multiple enzymes are used together, pH compatibility is an important factor for the enzymatic reactions. Actually, the pH and buffer concentration of reaction conditions were 8.0 and 500 mM, which were optimal for SH. But they are not optimal conditions for HsXyIDH2 and NOX. From the previous publications, the optimal pH of most NADH oxidases falls into the pH range of 5.0-6.0. The evaluation of NOX under the optimal conditions of SH is not fair to NOX.

In fact, the main reason to run the cascade at pH 8 was that most of the other enzymes especially the dehydrates have an optimum activity at alkaline pH 8-9 (ref. 41, main text) In addition, the best yield of 2-ketoglutarate with NOX were reported at 30°C, in Tris-HCl pH 8. We do agree with the reviewer that NOX by itself operate best at pH 7 and retained 70% of activity at pH 8. We have addressed now these issues by running temperature, buffer and pH screening for both system with HsXyIDH2. We added the new data to supporting information.

- 3) The specific activities of the NOX and SH are 50.1 U mg⁻¹ in 50 mM Tris-HCl pH 7.0 at 37°C and 1.5 U mg⁻¹ in Tris-HCl pH 8 at 22 °C, respectively. SH shows a quite low activity, which means the low catalytic efficiency of the enzyme. However, the results showed that total turnover number of SH was much higher than that of NOX. Please give more explanation in the text.

We do address this in the main text by showing that despite the low activity of SH (since it's the reverse reaction) when equi-molar amounts of both enzymes are compared, SH outperformed NOX in all setup because releasing H₂ is more efficient than dissolving O₂ in water. We do agree with the reviewer about the specific activities, however these data are enzyme characterization data, where the initial rates and kinetics are measured and not the total product formation. We actually thought about putting more emphasis on that in the main text in the first version. We added now a sentence at the beginning of the paragraph (line 171) to put more emphasis on this.

- 4) In the experiments, equi-unit enzymes of NOX and SH were used for the biocatalysis. The protein concentration of the enzyme will be highly different, and the protein concentration of SH will be 33-fold that of NOX in the reaction mixture. This might have some effects on the enzymatic reactions because of the low concentration of NOX.

This is correct, however the concentration of NOX was 4.5 μM this equals 3 U/mL or 1.5 U to each reaction of 0.5 ml -In the original draft it was mistakenly put as 1.5 U/mL not 1.5 U per reaction- it is now revised). 4.5 μM of NOX is already above the standard concentration of 1 μM that we used in the screening study.

SH was **not** 33-fold but just 4 fold, SH (19.6 μM), this is because SH has a molecular weights of approx. 200 kDa whereas NOX is approx. 50 kDa. There are also differences in the specific activities of both enzymes depending on enzyme preparation. The specific activities differ

between (1.5-0,7 U/mg) for SH and (30-15 U/mg) for NOX depending on the purification and FAD loading in case of NOX.

To solve the confusion we also added the concentration in μM to this graph.

- 5) The total turnover number of NOX from *Lactobacillus pentosus* is $6.8 \cdot 10^5$ in the previous reports. It is totally different to the data in this work. Please give the possible reasons.

This is true, however, in Nowak et al. the activity of NOX was directly measured but not the product formation after the biotransformation. The residual activity of NOX was measured with 0.3 mM NADH after incubating the enzyme at 37°C for a certain period of time. The data of each time point was then fitted and the deactivation constant (k_{deact}) was calculated which was eventually used to calculate TTN as $k_{\text{cat}}/k_{\text{deact}}$. The described TTN in Nowak et al. refers actually to the thermal stability of NOX rather than the total performance of the catalyst in whole the reaction and under the reaction conditions. Important parameters such as the stability of the cofactor after recycling and effect of the reaction conditions are unconsidered with in such analysis.

- 6) It might be better to add a description of SH stability in utilized conditions for comparison because the H_2 in the reaction mixture might have negative effects on SH and other enzymes.

This is a good point, in Herr et al (ref. 44) the stability of SH was fully studied on parameters like temperature, solvents and stirring. We agree with the reviewer that if H_2 would have been produced in high rates that might cause bubbles formation, which will result in the formation of gas-liquid interfaces and eventually deactivation of enzymes. At our scale and 1.5 U/mg of SH activity, the amount of H_2 produced are minimal to cause any harm. We could do not observe any gas bubbles forming in our reaction. Our only indicator for the H_2 formation was the data from the sensor. Temperature and stirring –especially the later- are the main reasons standing behind destabilizing the enzymes. Nevertheless, one has to stress that SH is not a stable enzyme, but neither is NOX, in fact both of enzymes share similar stability profiles something that need to be solved in upcoming studies for example through enzyme engineering of both enzymes.

- 7) It is noted that the enzymatic reactions were carried out in the close and open systems. The yield in the open system was 45% higher than that of the close one. Thus, it is necessary to evaluate the effects of the pressure and the solubility of H_2 in the system.

This a very good point, figures S5 and S6 show that the diffusion of H_2 in the open system is generally much faster. One will expect that in the closed system, the partial pressure of H_2 will increase which will cause a shift in the equilibrium constant and a decrease in the diffusion of H_2 from the liquid phase. Therefore, the solubility of H_2 will increase (indicated by the slow decrease in H_2 concentration (figure S5)) and the reaction will shift towards a H_2 -driven NAD^+ reduction (the thermodynamically favored reaction). This explain why the closed system has lower yields.

Concerning the effect of pressure, we so do see a slight increase in the pressure for about 17.9 mbar in 20 hours in the closed system; this seems minimal to cause any effect especially with 1 mL of a headspace, in the open system a slight decrease of 4 mbar was observed most probably due to slight temperature fluctuations.

Reviewer #3 (Remarks to the Author):

The authors state the thermodynamically beneficial irreversibility of the NAD⁺ regeneration reaction, which I agree with. However, so far the authors have used aldehyde to acid oxidation reactions, which intrinsically are irreversible themselves. It would be very interesting to see if the hydrogenase-system can also yield full conversion in challenging reaction systems such as the oxidation of primary or secondary alcohols to the corresponding carbonyl products. I am thinking about simple oxidations such as cyclohexanol or benzyl alcohol, which are typically difficult to drive to completion using substrate-coupled regeneration approaches.

We thank the reviewer very much for his valuable suggestion. We were very interested to test his idea. Following his suggestion, we performed a biocatalytic oxidation of alcohol to aldehyde.

Here, we have chosen the conversion of benzyl alcohol to benzaldehyde as our model reaction (Scheme below). The key enzyme here is in the BsADH form bacillus subtilis. We have cloned the gene, expressed in *E. coli* and purified the enzyme. The equilibrium of the reaction lies heavily on the reduction of aldehyde, i.e. the K_{eq} of oxidation reaction is as low as 4.6×10^{-5} .

We have tested 2 different combinations (1 mM NADH with 50 mM of substrate, 10 μ M of each enzyme) and (10 mM NADH with 100 mM substrate, with 20 μ M of each enzyme). The reaction were performed in 0.5 mL scale in Tris-HCl pH 8 at 30 °C and 300 rpm in pierced tubes.

SH did also outperformed NOX in these reactions as well, by driving the reaction towards alcohol oxidation by releasing H₂ to keep the cofactor oxidized (table below). The cofactor concentration played a crucial role here, unlike the sugar cascade increasing the NADH concentration lowered to total yield significantly, due the high K_{eq} of the reduction reaction of ADH reactions.

In none of the reactions full conversion was observed, which might not be caused only by the low equilibrium of the reaction but also might be related to the higher activity and stability of BsADH compared to both SH and NOX. Tuning this system towards achieving full conversion will require both optimizing the NADH and enzymes concentration plus optimizing the process itself i.e. exploring the effect of spiking of enzyme and immobilization. This is going to be the focus of upcoming studies.

We that think that these data might be interesting for this paper. However, since no full conversion could be achieved we would like to leave it for the reviewer to decide whether, we should include these results in the main text or wait until we study the system in more depth and optimize it, which will do for sure in the near future- and then address this in another paper.

Enzyme	NADH (mM)	Substrate (mM)	Yield (mM)	Conversion (%)
SH	1	50	23 ± 2.5	49
	10	100	14.4 ± 0.5	14
NOX	1	50	13.1 ± 0.4	30
	10	100	2.2 ± 0.1	2.5

Minor issues:

Line 36ff: The authors state that there are few effective NAD⁺ regeneration systems available and later (arbitrarily) mention some older systems. One could argue here that other NAD(P)⁺ regeneration systems have been reported in the past years. I am also not a big fan of the term 'efficient' per se if it is not quantified (TNs, TFs, STY etc.) and compared.

We agree with the reviewer, it has been revised

Line 70: I am not convinced that the hydrogenase-cased oxidation system is '100% atom efficient'. This is the case if reduction reactions are considered (then all atoms of the starting material are incorporated into the product). Here, H₂ as by-product is formed, hence the atom efficiency (as defined by Trost) is not perfect.

Atom efficiency here is referred to the native reaction of SH. i.e the H₂-driven reduction of NAD⁺ to NADPH. The atoms and electron of H₂ are transferred to NAD⁺. We added now "H₂-driven" to indicate this.

Line 78: I would not call it NADH oxidase but rather NADH oxidation activity. The term oxidase sounds too much like an O₂-dependent reaction.

We totally agree with the reviewer, has been revised

Line 92: The authors state that no H₂O₂ was detectable but it remains unclear why. Were these reactions performed anaerobically? Or was there some catalase-activity present in the enzyme preparation? Any other explanation?

The reactions were performed in micro titer plate, with 0.1 FMN and without shaking; this might not be the best condition to ensure O₂ transfer. In addition, SH can degrade O₂ to superoxide and water at the diaphorase domain. Furthermore, the FMN concentration plays a significant role here in the production of H₂O₂. Previous studies showed that for a considerable amount of H₂O₂, one would need to add 1mM of FMN. Echoing the reviewer, one cannot for sure completely exclude the presence of catalase in the enzyme preparations, since the expression of cytoplasmic catalases and peroxidases is upregulated in *Ralstonia eutropha* during the aerobic cultivation and productions of hydrogenases.

Line 116: 'The conversion rate of the substrate decreased drastically after the first minute...': As far as I can see the very high initial product formation rate is obtained from the first data point after one minute. I do not see a zero-value. So could the very high initial product formation rate be an artefact from this first value (maybe the t=0-value would not be zero?) If this is not the reason, the authors should provide some explanation for this observation.

This is an important point. This is correct. Our procedure was as following: for time point (1 min)

- 1- Start the reaction by putting enzymes into the 1. Reaction.
- 2- Incubation (1 min). This is the time needed to add the enzymes to all parallel reactions.
- 3- Taking samples for each reaction beginning from 1. Reaction, dilution and transferring the sample to the centrifuge (this all takes a 1 min).
- 4- Filtration (3 min)).

The same for every time point.

This also explains why there was no sampling for time point ($t = 0$ min), and the absence of $t = 3$. Since the reaction was too fast. Our $t=0$ samples of the other reactions (cascades etc.) showed already a product formation. In addition, it is very important to mention that filtration cannot stop the reaction immediately as by adding a strong acid or base. However, unfortunately, using strong acids/base is not compatible with our HPLC methods.

Should we do any action / add anything in this regard in the manuscript?

Figure 1: Please provide molar concentrations of all reagents (here of the enzymes). This counts for all figures and tables. Please use colour to make identification of the individual time course easier.

This has been revised we added the concentrations of all reagents to the experimental part. Our goal behind choosing the color was to keep the colors for SH (black) and NOX (gray) throughout the whole manuscript the same to make the comparison easier.

Line 118: $10 \text{ } \mu\text{mol}\cdot\text{h}^{-1}$: I assume the authors mean concentrations rather than absolute amounts here.

No, it was intended to be as a molar flow. Up to our knowledge, production rates are usually depicted in literature as a mass flow rates (g or mol per time) and not as a concentration rate. If the reviewer sees this otherwise we will be happy to revise.

Line 128: rather than 'thermodynamically unfavorable' better 'endergonic'. Thermodynamic feasibility consists of enthalpic and entropic terms.

This has been revised

Table 1: better add the enzyme concentrations in the table rather than using the complicated +, ++, +++ scheme. Conversion column: this is a bit unclear. Maybe use conversion and D-xylosate/a-KG-selectivity or display the concentrations of the products (preferred).

This has been revised. We would rather use conversion in % to avoid any misunderstandings. Due to evaporation, we measure a product concentration in the overnight samples of the open system – i.e. in case of full conversion- that are higher than the initial substrate added. We do correct these values by looking at the evaporation of the negative controls and the increase in the concentrations of the other components like (Tris, FMN, and glycerol) in all samples, which should remain unchanged after the reaction.

REVIEWER COMMENTS

Reviewer #1 (Remarks to the Author):

My comments have been addressed in the revised version of the manuscript. I am still not overly excited about this study, but the authors provide the necessary minimum of data now, and there is no objective reason to not publish the manuscript. I understand this is in the interest of the editor.

Reviewer #2 (Remarks to the Author):

I've gone through the manuscript. The authors have addressed most of the questions compared to the previous version. However, I have a doubt about the data in Figure 5S:
According to this figure, the total turnover number of NOX was about one-third of SH. But the conversion was much higher than that of SH. Please give some explanation.

Reviewer #3 (Remarks to the Author):

I am satisfied with the changes made.

Point to point

We thank all the reviewers for their feedback.

Response to Reviewer 2:

We presume the reviewer meant Figure 4S. We looked again into our raw data and updated the values, but still the same was observed. In Figure 4S the yield of SH and NOX are almost identical, when considering the error, which was caused by the sensor in one sample (we state this in the caption). It is correct that the TTN of SH is around 3 times higher than NOX because less SH was used (this is also indicated in the caption of Figure 4S). Figure 4S describes the effect of stirring on both systems; here the same amounts in mg of both enzymes (320 mg) were used. However, since the molecular weight of SH is 173 kDa and NOX is 51 kDa, in calculation of TTN = (mol product/mol active sites). The TTN of SH becomes approx. 3X higher than NOX.

REVIEWERS' COMMENTS

Reviewer #2 (Remarks to the Author):

The manuscript could be accepted in the present form.